# The mediating role of depression on the link between physical activity and health-related quality of life among people with diabetes: A cross-sectional study

**Djoko Priyono** [1,2], **Sanghee Kim** [1] *

1 College of Nursing, Keimyung University, Daegu, Republic of Korea, 2 School of Nursing, Faculty of Medicine, Tanjungpura University, Kota Pontianak, Indonesia

* shkim07@gw.kmu.ac.kr

## Abstract

### Introduction

A correlation between health-related quality of life (HRQoL) and physical activity has been identified. Many studies have discussed whether this correlation is significantly associated with depression in the general and diabetic populations. However, the role of depression in this relationship, especially in individuals with diabetes, remains incompletely understood.

### Objective

This study investigated the relationship between PA and HRQoL, with depression as a potential mediator, in individuals with diabetes.

### Methods

This cross-sectional study involved 1,472 individuals with diabetes who participated in the Korea Health Panel Survey (KHPS) from 2019 to 2020. Their sociodemographic characteristics, PA, depressive symptoms, and HRQoL based on EuroQol-five-dimension (EQ-5D) scores were extracted from the KHPS. The mediating effect of depression on PA and HRQoL was investigated using multiple regression and a mediation effect test.

### Results

HRQoL was positively associated with PA, regular exercise, and varying degrees of walking activity. Conversely, depression was substantially negatively associated with HRQoL. Mediation analysis confirmed that depression partially mediated the relationship between PA and HRQoL. Specifically, for PA and regular exercise, the indirect effect of depression accounted for 46.61% ($B = 0.002$, $p < 0.05$) and 33.82% ($B = 0.003$, $p < 0.001$).

**Data Availability Statement:** The raw data supporting this study's findings used version 2.2 dataset and are publicly available from the Korea Health Panel Survey repository with permission.

However, data availability for processed data is limited because the free sharing of data between individuals is prohibited according to the laws of the Korea Health Panel Survey. Interested researchers can replicate this study and may obtain the raw data by directly requesting the National Health Insurance Services and the Korea Institute for Health and Social Affairs. To access the data, interested researchers can download and complete the 'Korea Health Panel Survey Data User Agreement Form' (https://www.khp.re.kr:444/eng/data/data.do), select the desired data version, and send the form to the administrator of KHPS via email (khp@kihasa.re.kr) for consideration.

**Funding:** The author(s) received no specific funding for this work.

**Competing interests:** The authors have declared that no competing interests exist.

**Abbreviations:** BMI, Body mass index; EQ-5D, EuroQol-five dimension; GDS, Geriatric Depression Scale; HRQoL, Health-related quality of life; KHPS, Korea Health Panel Survey; PA, Physical activity.

## Conclusion

In individuals with diabetes, depression was found to mediate the effect of PA on HRQoL. Therefore, conducting depression screening and managing depressive symptoms in this population is crucial to enhancing HRQoL through PA interventions. Consequently, strategies to enhance HRQoL can be effectively implemented and customized in response to particular depression screening outcomes.

## Introduction

Regular physical activity (PA) is crucial for those with diabetes. It is widely acknowledged to aid in managing diabetes by enhancing glycemic control, mitigating the risks of diabetes-related complications, and promoting overall well-being, encompassing both mental and physical aspects [1, 2]. However, compared to the general population, those with diabetes tend to engage in less PA [3]. Furthermore, 40% of those with diabetes do not participate in sufficient physical activity based on the World Health Organization recommendations [4]. One study using the UK Biobank, which comprises over 233,000 participants, revealed that those with cardiometabolic disorders, including type 2 diabetes, reported significantly less PA, particularly vigorous PA, than healthy adults. This study also reported that 16% of the no-disease group had low PA levels, which increased with the onset of cardiometabolic disease. A lack of PA has also been linked to several risk factors, including multiple comorbidities, being overweight or obese, high stress, and lower health-related quality of life (HRQoL) [5, 6].

HRQoL refers to the impact of diabetes and its management on an individual's overall well-being, including physical, mental, and emotional health [7]. While several studies have examined the relationship between PA and HRQoL in the general population, few have focused on individuals with chronic diseases such as diabetes. Therefore, further research in this area is needed [8]. Furthermore, the sample sizes of many studies on individuals with type 2 diabetes have been relatively small, limiting the statistical power and generalizability of their findings. Furthermore, studies have not consistently accounted for important confounding variables, such as diabetes complications, comorbidities, and socioeconomic status, which may independently influence PA and HRQoL [9, 10]. Lastly, there is a lack of research investigating potential mediators, such as depression, with individuals with diabetes two- to three-fold more likely to experience depression [11, 12].

A more comprehensive examination of the processes involved in the relationship between independent and dependent variables can lead to new insights into the timing and nature of their influence. In this study, we considered depression as a mediator in the relationship between PA and HRQoL. Research has revealed that moderate PA levels and leisure activities are related to depression in patients with diabetes [13]. A previous longitudinal study found that moderate-intensity PA was associated with a 28% lower risk of developing major depression [14]. Conversely, depression negatively impacted HRQoL in patients with diabetes [15]. Several studies have shown depression to have a partial mediating effect on the relationships between HRQoL and neuropathic pain, atrial fibrillation, and cardiovascular disease. However, further investigation is needed in the case of patients with diabetes [16–18].

Based on our analysis, we propose possible mediating effects of depression on the relationship between PA and HRQoL. It is crucial to investigate the association between PA and HRQoL and the potential impact of depression on this relationship using data-driven inferences. This study had two primary aims: (i) to assess the relationship between PA and HRQoL

among individuals with diabetes mellitus in South Korea using a comprehensive, nationally representative database (Korea Health Panel Survey [KHPS]), and (ii) to explore depression as a potential mediator of the association between PA and HRQoL.

## Methods

### Study design

This study obtained data from the 2nd Korea Health Panel Survey (KHPS) version 2.2 dataset. The KHPS was undertaken collaboratively by the National Health Insurance Corporation and the Korea Institute for Health and Social Affairs. The data presented in this study is provided at the request of the author and is not disclosed for privacy purposes, https://www.khp.re.kr:444/eng/data/data.do (accessed on August 10, 2023). It is a nationwide public database comprised of households and their members. Each household and member are given a unique, confidential identification number protected by the data-gathering system. The data collection system and databases do not link the identification numbers to personal information to protect respondents' confidentiality. Trained interviewers conduct computer-assisted personal interviews with study participants in their homes.

The KHPS was separated into three study types: individual, household, and disease surveys. The household survey included questions about demographic characteristics, household premiums, living expenses, drug purchases, and private health insurance. The data collection process began with authors downloading and completing the data use agreement. Next, the respondent sent the completed form to KHPS and awaited approval and an email from KHPS. After approval and receiving the data by email, the KHPS extracted the data using the SPSS program. We only utilized open data from the KHPS and excluded any personal identification information. The study sample was derived from respondents who self-reported being diagnosed with diabetes mellitus. These individuals were asked whether they had chronic diabetes, to which they could respond yes or no. The inclusion and exclusion criteria were as follows. In 2020, respondents with diabetes mellitus ($n = 1,543$) were selected from among all KHPS respondents ($N = 14,844$). Then, those who did not report their PA levels ($n = 1,503$) or provide information about their alcohol consumption ($n = 1,472$) were excluded from the analysis by considering that non-response could be related to factors relevant to the study outcomes, potentially biasing the results. After excluding those with incomplete or missing information, the analysis comprised 1,472 respondents with diabetes mellitus (**Fig 1**).

### Measures

**Physical activity.**    We used the International Physical Activity Questionnaire short form (IPAQ-SF) to assess PA duration and frequency. The respondents were asked two key questions: "Over the past week, how many days did you participate in vigorous or moderate-intensity physical activities that caused slight breathlessness and an elevated heart rate for a minimum of 10 consecutive minutes?" and "On those days when you engaged in vigorous or moderate PA, approximately how many minutes did you typically dedicate to such activities?" The total time spent in PA was then calculated as the average number of days in the previous week where the respondent engaged in at least 20 minutes of vigorous-intensity PA or at least 30 minutes of moderate-intensity PA [19].

We also assessed the respondents' regular exercise and walking activity. Regular exercise was evaluated using the question, "Over the past year, have you exercised regularly?" The possible answers were "no exercise" and "regular exercise." Regardless of the type or degree of intensity, the categorization of PA into regular and irregular groups was based only on the frequency of engagement in PA. Finally, walking activity was evaluated using the question, "How

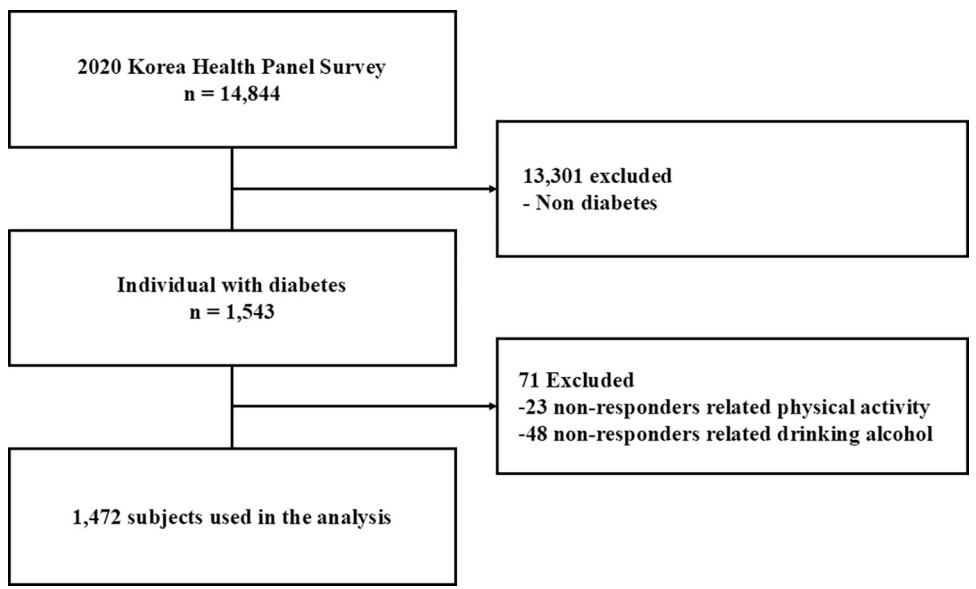

**Fig 1. Flow diagram displaying the inclusion and exclusion of subject.**

many days did you walk for at least ten minutes at a time throughout the last week?" The walking variables were categorized into three groups based on the seven-category answer options: "none," "≤3 days/week," and ">3 days/week."

**Depressive symptoms.** Depressive symptoms were assessed in the KHPS through a single question for screening depression, where this single question was validated against a diagnostic interview in the Yale Task Force in Geriatric, in which an overall 95% confidence interval indicated that the percent of subjects correctly identified by the depression question was unlikely to be more than 6.1% less than that identified by the Geriatric Depression Scale (GDS) [20]. A Single question was followed: "In the past year, have you experienced sadness or unhappiness for more than two weeks that disrupted your daily life?" The possible responses were "yes," indicating the presence of depressive symptoms, and "no," indicating the absence of depressive symptoms.

*HRQoL.* HRQoL has been assessed in the KHPS using the EuroQol-five dimension (EQ-5D) instrument since 2009 and is presently the subject of investigation. The KHPS examines several HRQoL factors, such as motor skills, daily activities, self-management, pain/discomfort, and anxiety/depression. Each question that described the respondent's current situation was assigned a point value: 1 = no problem, 2 = some problem, and 3 = extra problem. Higher scores indicated worse HRQoL. The EQ-5D index was analyzed based on the South Korean Time Trade-Off Values for EQ-5D health states as follows: EQ-5D index = 1 − (0.05 + 0.096×[Mobility = 2] + 0.418*[Mobility = 3] + 0.046×[Self-Care = 2] + 0.136×[Self-Care = 3] + 0.051×[Usual Activity = 2] + 0.208×[Usual Activity = 3] + 0.037×[Pain/Discomfort = 2] + 0.151×[Pain/Discomfort = 3] + 0.043×[Anxiety/Depression = 2] + 0.158×[Anxiety/Depression = 3] + 0.05×N3, where N3 is an interaction term. A dimension is classified as 1 if it belongs to level 2 or 3 and 0 otherwise. The weighted score was 1 when all five aspects of EQ-5D were rated as 1. A higher EQ-5D score indicates better HRQoL [21].

*Covariates.* The following variables were included in the analysis: smoking, body mass index (BMI), drinking habits, high stress, anxiety symptoms, suicidal thoughts, age, education level, economic situation, family composition, sex (male and female), and subjective health status. The analysis comprised individuals aged 20–94 years, who were divided into four age

groups: 20–29, 30–49, 60–64, and $\geq$65. Age was examined using the mean and standard deviation (SD). Education level was determined based on the KHPS level, which was then graded using a six-point Likert scale and restructured into dichotomous categories based on the study by Kim et al. as "employed" or "unemployed." [22]. Family compositions were divided into "living alone" and "living with others" based on the question, "Do you currently live in this household together?" Disability status was dichotomized into disabled and non-disabled. The BMI was calculated from the respondents' height and weight using the following formula: weight (kg)/height (m$^2$). According to the Korean Society for the Study of Obesity, respondents were considered obese if they had a BMI of $\geq$25 kg/m$^2$.

Smoking status was categorized into three groups: never-smoker, ex-smoker, and current smoker. Following Han et al. (2023), respondents who stated that they had smoked in the last 30 days were categorized as current smokers, and those who have abstained from smoking for one year or more are classified as ex-smokers. Alcohol use was categorized into three groups depending on the quantity and frequency of drinking: non-drinkers, moderate drinkers, and heavy drinkers. The KHPS question asked respondents to rate the stress they typically encounter daily as 1 (high stress levels), 2 (I often experience high stress levels), 3 (low stress levels), and 4 (no stress). The respondents provided ratings indicating their stress level, with two possible answers: yes (indicating a high stress level) and no (indicating no stress).

Anxiety symptoms were assessed by asking respondents about their experiences with excessive anxiety or worry that interfered with their daily lives for at least six months. The responses indicating the presence of anxiety symptoms were categorized as yes and no. We also included the covariate suicidal thoughts, which were assessed using the question, "Have you ever thought about dying in the past year?" and responses were categorized as yes and no. This variable was included with the consideration that suicidal ideation can substantially affect an individual's engagement in physical activity by reducing motivation, leading to social withdrawal, depleting energy levels and also profoundly influence various aspects of HRQoL, including mental health, physical functioning, and social relationships [23]. Finally, respondents' subjective health status was classified as "poor" or "fair/good" based on their general health descriptions.

## Statistical analysis

The data were analyzed using the SPSS software (version 25.0; IBM, Armonk, NY, USA) and JAMOVI (version 2.5.3). Categorical data are described using frequencies (percentages), and continuous data are described using means (SDs). Simple linear regression analyses based on the HRQoL measure were conducted to obtain unadjusted variable values. Then, multiple linear regression analyses were performed using the entered model to adjust each variable for HRQoL. Mediation analyses were conducted using the bootstrapping procedures MEDMOD package in JAMOVI to assess direct and indirect effects. In this study, we used three mediation models to investigate the role of depression as a mediating variable between various independent and dependent variables. The first model, which serves as the core of our analysis, examines the relationship between PA as the independent variable, depression as the mediating variable, and HRQoL as the dependent variable. The second model explores the mediating effect of depression in the context of regular exercise as the independent variable and HRQoL as the dependent variable. Finally, the third model investigates the mediating role of depression in the relationship between walking activity as the independent variable and HRQoL as the dependent variable. A two-tailed $p$-value of <0.05 was considered statistically significant for all analyses.

## Ethical statements

This study was conducted in accordance with the Declaration of Helsinki. The Korean Health Panel data used in this study is statistical data designated by the government pursuant to Article 18 of the Statistics Act. The contents of the questionnaire and survey results are only used for statistical purposes in establishing national policy. This study was exempt from ethical review and approval because only de-identified data was provided and used in accordance with the Personal Information Protection Act and Statistics Act to prevent individuals from being inferred from survey data. In addition, all personal information cannot be used to identify individuals and is fully protected in accordance with Articles 33 and 34 of the Statistics Act. Before participating in the KHPS, respondents were asked to read and sign an agreement form, which consented to use their data in future scientific research. All participants in this study received sufficient explanation about how to use the questionnaire, provided written consent, and then completed the questionnaire. This study does not include minors.

## Results

### Participants' characteristics

The study sample exhibited a relatively balanced sex distribution, comprising 49% males and 51% females. Most respondents were aged 60 to 64 years, lived with others, were employed, and had no disabilities. The proportion of respondents who engage in regular exercise is equal to the proportion of those who do not. Additionally, more than half of the respondents (53.5%) reported walking for at least three days per week. Light PA (40.6%) was more common than moderate and vigorous PA. Of the 1,472 respondents in the study sample, 27.3% experienced high stress levels, 11.8% exhibited depression symptoms, 8.1% reported anxiety, and 7.4% had suicidal thoughts. Furthermore, most respondents (44.4%) rated their subjective health status as somewhat. Their mean HRQoL score was 0.878 (**Table 1**).

Fig 2 illustrates the percentage of respondents who reported issues in the five EQ-5D dimensions, disaggregated by sex. More men than women reported having no problems in each dimension: Mobility (37.8% vs. 28%), Self-Care (44.8% vs. 43.5%), Usual Activities (41% vs. 34.9%), Pain/Discomfort (29.5% vs. 17.8%), and Anxiety/Depression (42% vs. 39.3%). The most common concern for both sexes was the Pain/Discomfort dimension, while the least common was the Self-Care dimension (**Fig 2**).

The factors that were initially considered to have an impact on HRQoL, such as sex, age >65 years, smoking status, drinking status, high stress, and anxiety symptoms, were found to not be significant after adjustment. In contrast, economic status, disability, and subjective health status consistently influenced HRQoL before and after adjustment ($p < 0.001$). The primary variable in this study was HRQoL, which showed a negative association with depression ($B = -0.051$, 95% confidence interval [CI] = $-0.026$–$-0.076$) and positive associations with regular exercise, the three types of walking status, and PA ($p < 0.001$) (**Table 2**).

Tables **2** and **3** reveal several significant associations and mediating effects. HRQoL exhibited a negative relationship with depression and positive associations with regular exercise, varying levels of walking activity, and different PA intensities ($p < 0.001$). The associations between HRQoL and regular exercise, walking, and PA were found to be mediated by depression. Specifically, the indirect effect of depression accounted for 33.82% ($B = -0.003$, $p < 0.001$), 46.61% ($B = 0.002$, $p < 0.05$), and 13.86% ($B = 0.003$, $p < 0.05$) of the effect of regular exercise, PA, and walking on HRQoL, respectively. Fig 3 shows the interrelationships among PA, walking, regular exercise, depression, and HRQoL. These results indicate that

**Table 1. Characteristics of the sample.**

| Characteristics | N = 1472 | Characteristics | N = 1472 |
|---|---|---|---|
| Gender, n (%) | | BMI status, n (%) | |
| Male | 721(49.0) | Non obese | 896 (60.9) |
| Female | 751(51.0) | Obese | 576 (39.1) |
| Age (years), Mean (SD) | 68.8 (10.7) | Smoking status, n (%) | |
| Family Composition, n (%) | | Ex Smoker | 427 (29.0) |
| Live Together | 1459 (99.1) | Current smoker | 221 (15.0) |
| Living alone | 13 (0.9) | Never smoking | 824 (56.0) |
| Diabetes age, n (%) | | Drinking status, n (%) | |
| 20–29 years | 10 (0.7) | Non-Drinker | 819 (55.6) |
| 30–49 years | 331 (22.5) | Moderate drinker | 401 (27.2) |
| 60–64 years | 761 (51.7) | Heavy drinker | 252 (17.1) |
| 65 years older | 370 (25.1) | | |
| Educational Status, n (%) | | High Stress, n (%) | |
| Elementary School/ lower | 559 (38.0) | Yes | 402 (27.3) |
| Middle School | 281 (19.1) | No | 1070 (72.7) |
| High School | 400 (27.2) | | |
| College degree or above | 232 (15.8) | | |
| Economic Status, n (%) | | Depressive symptom, n (%) | |
| Employed | 758 (51.5) | Yes | 174 (11.8) |
| Unemployed | 714 (48.5) | No | 1298 (88.2) |
| Disability Status, n (%) | | Anxiety symptom, n (%) | |
| Disable | 211 (14.3) | Yes | 119 (8.1) |
| Non-Disable | 1261 (85.7) | No | 1352 (91.9) |
| Regular Exercise, n (%) | | Thought of suicide, n (%) | |
| Yes | 736 (50.0) | Yes | 109 (7.4) |
| No | 736 (50.0) | No | 1363 (92.6) |
| Walking Status, n (%) | | Subjective Health Status, n (%) | |
| < 3 Days/week | 329 (22.4) | Poor | 546 (37.1) |
| > 3 Days/week | 787 (53.5) | Fair | 653 (44.4) |
| | | Good | 273 (18.5) |
| Physical Activity, n (%) | | HRQoL, Mean (SD) | 0.878 (0.153) |
| Light | 597 (40.6) | | |
| Moderate | 449 (30.5) | | |
| Vigorous | 426 (28.9) | | |

depressive symptoms partially mediate the relationships between PA levels ($p<0.001$), active participation in regular exercise ($p<0.001$), walking frequency ($p<0.05$), and HRQoL (**Fig 3**).

## Discussion

Our study was the first to investigate the relationship between PA and HRQoL among individuals with diabetes mellitus in South Korea. Utilizing a large, nationally representative sample from the KHPS, our study provides novel insights into this association within the South Korean diabetic population. Our findings diverge from those of previous epidemiological studies, which have primarily focused on examining depression as a mediator between PA and HRQoL in elderly populations. Differences in sociodemographic characteristics and medical profiles between the elderly cohorts of past studies and the respondents in our study limit the

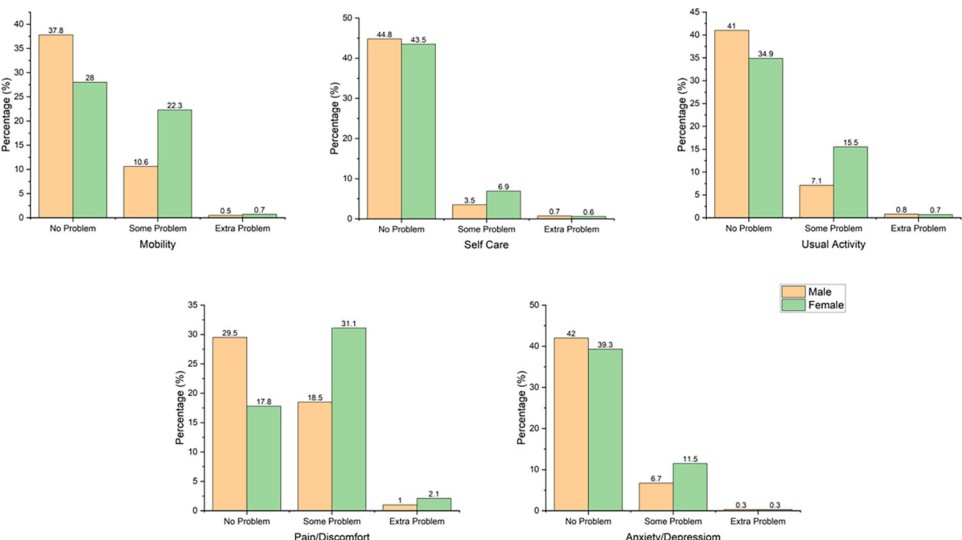

**Fig 2. Distribution of EuroQol 5-dimensions by sex.**

direct comparison of results. Our findings are also expected to contribute to developing targeted interventions to improve HRQoL among individuals with diabetes in South Korea. By elucidating the intricate relationships between PA, depression, and HRQoL within this population, our study provides valuable insights to inform the design and implementation of tailored strategies to enhance overall well-being in the diabetic population.

In our study, most individuals with diabetes performed regular exercise and light PA. A previous countrywide cross-sectional study in England showed that patients with diabetes exhibited no regular exercise, poor eating habits, high smoking, high BMI, high comorbidities, high stress levels, and poor HRQoL [4]. In contrast, individuals with diabetes in South Korea tended to smoke less, were non-obese, had lower perceived stress levels, and had an average total HRQoL score of 0.8 out of a maximum of 1, indicating no significant problems with their HRQoL.

Our study found that men had better HRQoL than women, consistent with other studies that found men with type 2 diabetes had a slightly greater overall HRQOL than women with type 2 diabetes [24, 25], who reported more diabetes-related worries and less ability. Men tend to have greater physical strength and muscle mass, which can contribute to better overall health and mobility. Psychologically, men often report lower levels of anxiety and depression compared to women, which can positively impact their overall sense of well-being. Additionally, societal expectations and gender roles play a significant part in shaping QOL perceptions. Men also typically face less pressure regarding appearance and weight management, which can lead to better body image and self-esteem. However, these differences in QOL are not universal or absolute, and individual experiences may vary significantly based on personal circumstances and cultural contexts [26]. Furthermore, HRQoL can be influenced by various factors, including psychosocial challenges (e.g., more significant diabetes-related distress and body image issues), biological factors (e.g., higher risk of certain complications), barriers to accessing quality diabetes care, and socioeconomic disparities (e.g., lower incomes and education levels) [27, 28].

Our findings align with previous research, indicating that greater PA and regular exercise are associated with lower risks and fewer symptoms of depression. This negative correlation between PA, exercise, and depressive symptoms is well-established in the literature. For

**Table 2. Relationships between participants' characteristics and HRQoL.**

| Subgroup | Unadjusted | | | Adjusted | |
|---|---|---|---|---|---|
| | B (95% CI) | *p*-value | | B (95% CI) | *p*-value |
| Gender | | | | | |
| Male | | | | | |
| Female | -0.060(-0.076 - -0.045) | < .001 | | 0.004(-0.025–0.017) | 0.707 |
| Diabetes age | | | | | |
| 20–29 years | | | | | |
| 30–49 years | -0.049(-0.144–0.045) | 0.306 | | 0.043(-0.119–0.032) | 0.259 |
| 60–64 years | -0.066(-0.160–0.027) | 0.136 | | 0.047(-0.123–0.028) | 0.219 |
| 65 years older | -0.124(-0.219 - -0.030) | 0.010 | | 0.068(-0.145–0.007) | 0.077 |
| Educational Status | | | | | |
| Elementary School/ lower | | | | | |
| Middle School | 0.062(0.041–0.083) | < .001 | | 0.030(0.013–0.048) | < .001 |
| High School | 0.095(0.076–0.114) | < .001 | | 0.044(0.028–0.061) | < .001 |
| College degree or above | 0.116(0.094–0.138) | < .001 | | 0.046(0.025–0.067) | < .001 |
| Family Composition | | | | | |
| Living alone | | | | | |
| Live Together | 0.015(-0.067–0.099) | 0.709 | | 0.020(-0.055–0.096) | 0.603 |
| Economic Status | | | | | |
| Unemployed | | | | | |
| Employed | -0.0896(-0.105 - -0.074) | < .001 | | 0.025(0.011–0.038) | < .001 |
| Disability Status | | | | | |
| Disable | | | | | |
| Non-Disable | 0.111(0.089–0.133) | < .001 | | 0.058(0.040–0.076) | < .001 |
| Regular Exercise | | | | | |
| No | | | | | |
| Yes | -0.066(-0.081 - -0.051) | < .001 | | 0.013(0.001–0.028) | < .001 |
| Walking Status | | | | | |
| None | | | | | |
| < 3 Days/week | 0.124(0.103–0.145) | < .001 | | 0.057(0.035–0.079) | < .001 |
| > 3 Days/week | 0.130(0.113–0.148) | < .001 | | 0.062(0.041–0.084) | < .001 |
| Physical Activity | | | | | |
| Light | | | | | |
| Moderate | 0.080(0.062–0.098) | < .001 | | 0.001(0.017–0.021) | < .001 |
| Vigorous | 0.101(0.083–0.119) | < .001 | | 0.009(0.011–0.029) | < .001 |
| BMI | | | | | |
| Non-Obese | | | | | |
| Obese | -0.007(-0.023–0.008) | 0.339 | | 0.004(0.012–0.013) | 0.920 |
| Smoking status | | | | | |
| Ex Smoker | | | | | |
| Current smoker | 0.024(-6.164–0.004) | 0.056 | | 0.006(0.013–0.026) | 0.534 |
| Never smoking | -0.033(-0.050 - -0.015) | < .001 | | 0.007(0.013–0.028) | 0.458 |
| Drinking status | | | | | |
| Non-Drinker | | | | | |
| Moderate drinker | 0.076(0.058–0.093) | < .001 | | 0.013(0.002–0.028) | 0.092 |
| Heavy drinker | 0.088(0.068–0.109) | < .001 | | 0.010(0.009–0.029) | 0.307 |
| High Stress | | | | | |
| No | | | | | |

(*Continued*)

**Table 2.** (Continued)

| Subgroup | Unadjusted | | | Adjusted | |
|---|---|---|---|---|---|
| | B (95% CI) | *p*-value | | B (95% CI) | *p*-value |
| Yes | -0.039(-0.057 - -0.022) | < .001 | | 0.013(0.001–0.028) | 0.077 |
| Depressive symptom | | | | | |
| No | | | | | |
| Yes | -0.127(-0.151 - -0.104) | < .001 | | -0.051(-0.026 –-0.076) | < .001 |
| Anxiety symptom | | | | | |
| No | | | | | |
| Yes | -0.125(-0.153 - -0.097) | < .001 | | 0.012(0.016–0.041) | 0.388 |
| Thought of suicide | | | | | |
| No | | | | | |
| Yes | -0.113(-0.143 - -0.084) | < .001 | | 0.028(0.002–0.054) | 0.030 |
| Subjective Health Status | | | | | |
| Poor | | | | | |
| Fair | 0.138(0.146–0.153) | < .001 | | 0.076(0.061–0.091) | < .001 |
| Good | 0.166(0.146–0.185) | < .001 | | 0.009(0.077–0.115) | < .001 |

example, a large-scale, cross-sectional, population-based study conducted across 36 low- and middle-income countries revealed a significantly higher prevalence of low PA among individuals with depression than those without depression (26.0% vs. 15.8%, $p < 0.0001$) [29]. Furthermore, systematic reviews and meta-analyses assessing the effects of PA on depression in individuals with type 2 diabetes mellitus have consistently demonstrated the effectiveness of PA in reducing the severity of depressive symptoms (standardized mean difference = −0.57, $p < 0.05$). These findings underscore the importance of promoting PA and regular exercise as potential strategies for mitigating the risk and alleviating the symptoms of depression, particularly among individuals with chronic conditions such as diabetes mellitus. By incorporating

**Table 3. The effects of PA on HRQoL through depression.**

| Variable | Mediated Effect | | |
|---|---|---|---|
| | Effect | 95% CI | (%) |
| Physical Activity | | | |
| Indirect effect | 0.002*** | -4.224–0.004 | 46.61 |
| Direct effect | 0.050*** | 0.041–0.059 | |
| Total Effect | 0.052*** | 0.043–0.061 | |
| Regular Exercise | | | |
| Indirect effect | 0.003*** | -0.008–1.094 | 33.82 |
| Direct effect | 0.062*** | -0.077 - -0.047 | |
| Total Effect | 0.066*** | -0.081 - -0.051 | |
| Walking Status | | | |
| Indirect effect | 0.003** | 0.001–0.006 | 13.86 |
| Direct effect | 0.055** | 0.047–0.064 | |
| Total Effect | 0.058** | 0.050–0.068 | |

CI: confidence interval.

**p<0.05.

***p<0.001

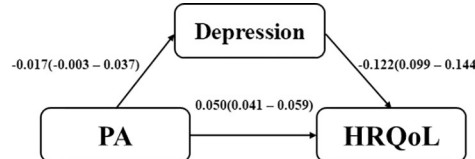

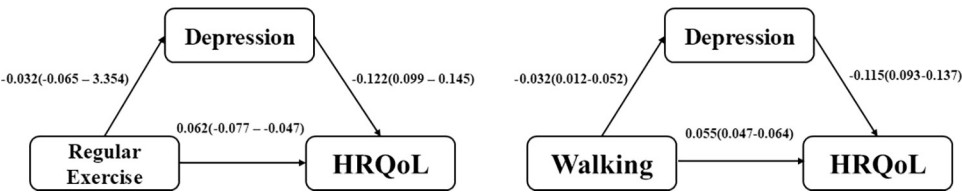

**Fig 3. Estimation of direct effects of physical activity, regular exercise, and walking on HRQoL through depression.**

regular PA into their lifestyles, they may experience improvements in their mental well-being and overall HRQoL.

PA has been found to alleviate depressive symptoms through various physiological pathways. One key mechanism involves the increased production of neurotrophic factors, such as brain-derived neurotrophic factor (BDNF), which promotes neuroplasticity and the generation of new neurons (neurogenesis) in the brain. These processes improve depressive symptoms by enhancing neural connectivity and facilitating the growth of new neural pathways [30]. Moreover, regular exercise plays a vital role in regulating the levels and activity of neurotransmitters such as serotonin, norepinephrine, and dopamine, which are crucial for maintaining a balanced mood state. By modulating these neurotransmitter systems, PA can help restore the proper functioning of mood regulation pathways in the brain [30]. Another important mechanism through which PA combats depression is normalizing the dysregulated activity of the hypothalamic-pituitary-adrenal axis. This axis controls how the body reacts to stress, and its dysregulation has been associated with the development of depressive disorders. The release of mood-enhancing hormones and growth factors such as endorphins, vascular endothelial growth factor (VEGF), insulin-like growth factor 1 (IGF1), oxytocin, and arginine vasopressin further contribute to the antidepressant effects of exercise [31, 32]. However, several studies have found this relationship to be bidirectional [33, 34].

Consistent with previous research, our study found that depression can influence HRQoL. Patients suffering from both depression and diabetes frequently reported greater difficulties with mobility and self-care activities and increased pain and discomfort than those with diabetes alone [35]. The effects of depression on HRQoL domains are likely due to the added burden of managing a chronic illness such as diabetes, the physiological mechanisms underlying both conditions, and the reduced motivation and ability to care for oneself properly. Addressing mental health should be a crucial element of comprehensive diabetes care to improve overall well-being in this population.

Our study showed that regular exercise, walking activity, and different types of PA are also positively associated with HRQoL. This finding is consistent with a previous which explored the relationship between PA patterns and QoL among individuals with type 2 diabetes mellitus in Ghana [36]. They revealed that PA, particularly walking and vigorous-intensity activities, was associated with better QoL outcomes. Another study also found that individuals who did

PA and regularly walked showed significantly better general health, vitality, and social functioning, which are part of HRQoL [37].

Depression significantly mediated the relationship between HRQoL and PA, indicating that PA influences HQoL directly and that depression acts as an indirect mediator. Patients with diabetes are at significantly greater risk of developing depression, with prevalences from 18% to 21% [38, 39]. While the mediation model accounted for a substantial portion of the variance in HRQoL, additional mediating factors may exist beyond depression. Depression may explain only one aspect of the relationship between PA and HRQoL. Further research is warranted to explore the potential role of other mediating variables that could contribute to this association.

Our findings emphasize the mediating effect of depression on the relationship between PA and HRQoL. They highlight the importance of mental health concerns in individuals with diabetes as part of a comprehensive approach to improving their overall well-being. They also emphasize the significance of developing and implementing targeted programs that assist in preventing or reducing depression among individuals with diabetes. In addition to promoting PA, healthcare providers and policymakers can work towards enhancing HRQoL outcomes for individuals with diabetes by addressing the mental health challenges faced by this patient population. For example, by encouraging patients with diabetes to do a personalized Physical Activity Package (PAP) recommending 150 minutes of moderate physical activity per week, such as brisk walking, swimming, or cycling. Encourage motivation and adherence by participating in group activities or connecting patients with diabetes support groups. Regular assessment of depression and anxiety symptoms allows for timely adjustments to the PA intervention and leveraging technology through text messaging interventions, mobile apps, or telehealth options can provide additional support between in-person visits.

Our study had several limitations that should be considered. First, this study used data from KHPS in 2020, where data were collected during the COVID-19 pandemic and associated restrictions, such as reduced physical activity due to lockdowns and increased mental health challenges stemming from isolation and anxiety, may have skewed results related to depression and overall quality of life. Second, its reliance on self-report scales for measurement may introduce potential for information bias since respondents' answers may be subject to recall errors or influenced by social desirability. Third, its cross-sectional research design precludes establishing a causal relationship between PA and HRQoL. Fourth, the fact that the tool employed to measure depressive symptoms only has one question with yes/no response options is another drawback. Future studies should use standardized instruments to support the emergence of depressive symptoms. Despite these limitations, our study had several strengths, including a large sample size of 1,472 individuals with diabetes. Furthermore, its analysis considered important confounding factors, enhancing the validity of the findings. Our results highlight the significance of considering mental health factors when examining the influences of PA on HRQoL outcomes in this patient population. Furthermore, they highlight the importance of developing comprehensive interventions that address both PA promotion and depression management, with the ultimate goal of optimizing the well-being of individuals with diabetes.

## Conclusions

In conclusion, HRQoL was significantly and positively correlated with PA, regular exercise, and differing degrees of walking activity. In contrast, HRQoL was significantly and negatively correlated with depression. The mediating effect of depression on the relationship between PA and HRQoL was confirmed in individuals with diabetes. It is imperative to conduct depression

screening and symptom management in this population, which will allow the implementation of strategies that effectively enhance HRQoL via PA interventions. Furthermore, future study should focus on longitudinal designs and interventional studies to more definitively determine the effects of physical activity on depression and HRQoL over time.

## Author Contributions

**Conceptualization:** Sanghee Kim.

**Data curation:** Djoko Priyono.

**Supervision:** Sanghee Kim.

**Writing – original draft:** Djoko Priyono.

**Writing – review & editing:** Sanghee Kim.

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
