## [Decision Letter · Decision Letter 0]

30 Jul 2024

PONE-D-24-24159The mediating role of depression on the link between physical activity and health-related quality of life among people with diabetes: A cross-sectional studyPLOS ONE

Dear Dr. Kim,

Thank you for submitting your manuscript to PLOS ONE. After careful consideration, we feel that it has merit but does not fully meet PLOS ONE’s publication criteria as it currently stands. Therefore, we invite you to submit a revised version of the manuscript that addresses the points raised during the review process.

We look forward to receiving your revised manuscript.

Kind regards,

Md. Feroz Kabir, BPT, MPT, MPH, BPED, MPED

Academic Editor

PLOS ONE

4. Please include a copy of Table 1, 2, and 3 which you refer to in your text on page 10.

Additional Editor Comments:

Please submit the revised paper as the direction of the reviewers within next 15 days.

Reviewers' comments:

Reviewer's Responses to Questions

**Comments to the Author**

1. Is the manuscript technically sound, and do the data support the conclusions?

Reviewer #1: Yes

Reviewer #2: Yes

2. Has the statistical analysis been performed appropriately and rigorously? 

Reviewer #1: Yes

Reviewer #2: Yes

3. Have the authors made all data underlying the findings in their manuscript fully available?

Reviewer #1: Yes

Reviewer #2: Yes

4. Is the manuscript presented in an intelligible fashion and written in standard English?

Reviewer #1: Yes

Reviewer #2: Yes

5. Review Comments to the Author

Reviewer #1: The study assessed the mediating effect of depression on the relationship between physical activity and health-related quality of life among people with diabetes. The study has added knowledge. Few comments for the authors.

Abstract

i. "decreased health-related quality of life (HRQoL) and physical activity". This phrase looks contradictory to what is known. You either remove the word 'decreased' or changed 'PA' to physical inactivity.

ii. "In individuals with diabetes, depression was found to moderate the effect of PA on HRQoL." Change moderate to mediate.

Introduction

i. " Lastly, there is a lack of research investigating potential mediators, such as depression, with individuals with diabetes two- to three-fold more likely to experience depression. The last part of the sentence needs referencing.

ii. "Several studies have shown depression to have a partial mediating effect on the relationships between HRQoL and

neuropathic pain, atrial fibrillation, and cardiovascular disease." Please reference 2-3 of these studies.

Statistical analysis

" These moderated mediation models included regular exercise, walking, and PA as independent variables, depressive symptoms as mediators, and HRQoL as the dependent variable." What variable serves as moderator(s)?

Results

i. Tables 1-3 are missing.

ii. Fig 3: It seems three sets of mediation analyses were performed. PA, Walking and RE were entered separately as an independent variable. Under the analysis section, you stated MEDMOD model was used suggesting walking and RE were moderators while depression was mediator. PA as independent and HRQoL as dependent variables. Please clarify.

Conclusion

There is a difference between mediator and moderator. Please be consistent in the usage. You set out to find out the mediating effect of depression on the relationship between PA and HRQoL. Please change moderating to mediating.

See other comments in the attached.

Reviewer #2: Well-written manuscript with with an adequate justification and good methodology. The conclusions were fair.

Minor corrections

Introduction

• Provide reference for line 40 (sentence starting with “However…..”)

Ethical consideration

Line 184: check the spelling of ‘providesd’

Results

Lines 196-197: The sentence starting with “Regarding PA levels….) must be rephrased to make the meaning clearer

6. PLOS authors have the option to publish the peer review history of their article (what does this mean?). If published, this will include your full peer review and any attached files.

Reviewer #1: No

Reviewer #2: No

---

## [Author Response · Author response to Decision Letter 0]

5 Aug 2024

Responses to Editor’s and Reviewers Comments

Manuscript ID: PONE-D-24-24159

Dear editorial staffs in Journal of PLOS ONE

I am truly grateful for your thorough consideration and precise review. I revised the manuscript according to your informative comments and suggestions. I sincerely hope that this revision has improved this manuscript to the level of satisfaction of the editor, reviewer and acceptance for publication, and I sincerely appreciate that the quality of this manuscript can be improved. 

Answers to specific comments, suggestions, and queries are as follows.

Response to Journal Requirements:

Comments 1: When submitting your revision, we need you to address these additional requirements. Please ensure that your manuscript meets PLOS ONE's style requirements, including those for file naming. 

Response 1: Thank you to the editor for the corrections. Following your suggestions, we have improved this manuscript according to the guidelines. We have submitted three new files with the following names based on your instructions: "Response to Reviewers," "Revised Manuscript with Track Changes," and "Manuscript." Additionally, we have updated the authors' affiliation titles, and the main body section based on the sample template provided by PLOS ONE.

Comments 2: We note that you have indicated that there are restrictions to data sharing for this study. PLOS only allows data to be available upon request if there are legal or ethical restrictions on sharing data publicly.

Response 2: Thank you very much for pointing this out. The data used in this study have data sharing restrictions. In the data availability statement section, we state that the data used in this study is wholly owned by the Korea Health Panel Survey. To obtain data in this study, the author has received approval from KHPS by first filling out the Korea Health Panel data Utilization consent form, which contains the following provisions: what version of data is desired, stating the name of the institution Korea Institute for Health and Social Affairs and the National Health Insurance Corporation in the research methods section, stating the data version, agreeing to use the data will be used only for research purposes and will not be used for personal or institutional commercial purposes, and the data will not be rented or transferred to others. We have included these points in the methods and data availability statement sections (Line 369-376).

Comments 3: Please amend your list of authors on the manuscript to ensure that each author is linked to an affiliation. Authors’ affiliations should reflect the institution where the work was done (if authors moved subsequently, you can also list the new affiliation stating, “current affiliation:.” as necessary).

Response 3: Thank you to the editor for the corrections. We have updated the authors' affiliation titles and main body section based on the sample template provided by PLOS ONE. (https://journals.plos.org/plosone/s/file?id=wjVg/PLOSOne_formatting_sample_main_body.pdf and https://journals.plos.org/plosone/s/file?id=ba62/PLOSOne_formatting_sample_title_authors_affiliations.pdf.)

Comment 4: Please include a copy of Table 1, 2, and 3 which you refer to in your text on page 10.

Response 4: Thank you for your nice reminder. We have added a data table into the main body manuscript with the following table position details as follow: Table 1 (Line 225), table 2 (line 241), table 3 (line 243).

Comments 5: Please review your reference list to ensure that it is complete and correct. If you have cited papers that have been retracted, please include the rationale for doing so in the manuscript text or remove these references and replace them with relevant current references. Any changes to the reference list should be mentioned in the rebuttal letter that accompanies your revised manuscript. If you need to cite a retracted article, indicate the article’s retracted status in the References list and also include a citation and full reference for the retraction notice.

Response 5: Thanks for your kind reminders. We have checked the completeness of the references and ensured that the references are written according to PLOS One guidelines.

Response to reviewer 1

Comment 1: Abstract

"Decreased health-related quality of life (HRQoL) and physical activity". This phrase looks contradictory to what is known. You either remove the word 'decreased' or changed 'PA' to physical inactivity.

Response 1: Thank you very much for the reminder. We have made revisions accordingly.

Comment 2: Abstract 

"In individuals with diabetes, depression was found to moderate the effect of PA on HRQoL." Change moderate to mediate.

Response 2: Thank you very much for the reminder. We have made revisions accordingly.

Comment 1: Introduction part 

" Lastly, there is a lack of research investigating potential mediators, such as depression, with individuals with diabetes two- to three-fold more likely to experience depression. The last part of the sentence needs referencing.

Response 1: Thank you very much for the reminder. We have made revisions accordingly.

Comment 2: Introduction part

"Several studies have shown depression to have a partial mediating effect on the relationships between HRQoL and neuropathic pain, atrial fibrillation, and cardiovascular disease." Please reference 2-3 of these studies.

Response 2: Thank you very much for the reminder. We have made revisions by adding 3 studies that support this statement.

Comment 3: Statistical analysis

" These moderated mediation models included regular exercise, walking, and PA as independent variables, depressive symptoms as mediators, and HRQoL as the dependent variable." What variable serves as moderator(s)?

Response 3: Thank you for the correction. We are sure that through your review, this article will be better. We apologize for the mistake because there is the word "mediator." From the beginning, we have only focused on the mediating effect of depression. For that, we have removed the word “mediator” and replaced it with the word mediating variable in this part.

Comment 4: Results part

Tables 1-3 are missing.

Response 4: Thank you very much for your previous comments that helped us improve this

manuscript. We have added a data table into the main body manuscript with the following table position details as follow: Table 1 (Line 225), table 2 (line 241), table 3 (line 243).

Comment 5: Results part

Fig 3: It seems three sets of mediation analyses were performed. PA, Walking and RE were entered separately as an independent variable. Under the analysis section, you stated MEDMOD model was used suggesting walking and RE were moderators while depression was mediator. PA as independent and HRQoL as dependent variables. Please clarify.

Response 5 Thank you very much for pointing this out. The core of our research analysis is to find out the mediating effect of depression on the relationship between PA as an independent variable and HRQoL as a dependent variable. However, we also want to see the mediating effect of depression as a mediating variable on the relationship between walking and regular exercise as independent variables and HRQoL as a dependent variable. For that, we created three mediating models. In the statistical analysis section, we revised the sentence as follows:

“In this study, we used three mediation models to investigate the role of depression as a mediating variable between various independent and dependent variables. The first model, which serves as the core of our analysis, examines the relationship between PA as the independent variable, depression as the mediating variable, and HRQoL as the dependent variable. The second model explores the mediating effect of depression in the context of regular exercise as the independent variable and HRQoL as the dependent variable. Finally, the third model investigates the mediating role of depression in the relationship between walking activity as the independent variable and HRQoL as the dependent variable. 

Comment 6: Conclusion part

There is a difference between mediator and moderator. Please be consistent in the usage. You set out to find out the mediating effect of depression on the relationship between PA and HRQoL. Please change moderating to mediating.

Response 6: Thank you very much for the reminder. We revised the sentence as follows: “The mediating effect of depression on the relationship between PA and HRQoL was confirmed in individuals with diabetes” (Line 353)

Response to reviewer 2

Comment 1: Introduction part

Provide reference for line 40 (sentence starting with “However…..”)

Response 1: Thank you very much for the reminder. We have added a reference to this sentence.

Comment 2: Ethical consideration part 

Line 184: check the spelling of ‘providesd’

Response 2: Thank you very much for the reminder. We have made revisions with change the word ‘providesd’ to be ‘provided’ (Line 206)

Comment 3: Results 

Lines 196-197: The sentence starting with “Regarding PA levels….) must be rephrased to make the meaning clearer

Response 3: Thank you very much for the reminder. We revised the sentence as follows: The proportion of respondents who engage in regular exercise is equal to the proportion of those who do not. (Line 218)

---

## [Decision Letter · Decision Letter 1]

14 Oct 2024

The mediating role of depression on the link between physical activity and health-related quality of life among people with diabetes: A cross-sectional study

PONE-D-24-24159R1

Dear Sanghee Kim,

We’re pleased to inform you that your manuscript has been judged scientifically suitable for publication and will be formally accepted for publication once it meets all outstanding technical requirements.

Kind regards,

Md. Feroz Kabir, BPT, MPT, MPH, BPED, MPED

Academic Editor

PLOS ONE

Additional Editor Comments (optional):

Submit your revised manuscript as the reviewers comments within the next 10 days.

Reviewers' comments:

Reviewer's Responses to Questions

**Comments to the Author**

1. If the authors have adequately addressed your comments raised in a previous round of review and you feel that this manuscript is now acceptable for publication, you may indicate that here to bypass the “Comments to the Author” section, enter your conflict of interest statement in the “Confidential to Editor” section, and submit your "Accept" recommendation.

Reviewer #1: All comments have been addressed

Reviewer #3: All comments have been addressed

Reviewer #4: (No Response)

2. Is the manuscript technically sound, and do the data support the conclusions?

Reviewer #1: Yes

Reviewer #3: Yes

Reviewer #4: Yes

3. Has the statistical analysis been performed appropriately and rigorously? 

Reviewer #1: Yes

Reviewer #3: Yes

Reviewer #4: No

4. Have the authors made all data underlying the findings in their manuscript fully available?

Reviewer #1: (No Response)

Reviewer #3: Yes

Reviewer #4: Yes

5. Is the manuscript presented in an intelligible fashion and written in standard English?

Reviewer #1: (No Response)

Reviewer #3: Yes

Reviewer #4: Yes

6. Review Comments to the Author

Reviewer #1: (No Response)

Reviewer #3: (No Response)

Reviewer #4: I belive the authors did great work but I have some comments that need to be addressed:

method:

The inclusion/exclusion criteria are well-defined, but the choice to exclude respondents based on missing data on PA and alcohol consumption could be more justified. For example, did the authors consider any potential bias from these exclusions?

The description of how depressive symptoms were measured ("sadness or unhappiness for more than two weeks") might benefit from elaboration. A discussion on the validation of this question as an adequate measure of depression could strengthen this section.

statistical analysis:

There is some ambiguity in the choice of covariates. While most of the covariates seem appropriate, variables like "suicidal thoughts" (Table 1) may require additional justification for inclusion. How might these impact PA and HRQoL? A brief explanation would be helpful.

In Table 3, the confidence intervals for the indirect effects appear quite narrow. It would be helpful to confirm if the bootstrapping method was used, as it’s typical in mediation analyses to ensure robustness of the estimates.

Results:

The sentence regarding the mediation analysis results (lines 250-251) could be made clearer. For example, explicitly stating how the interpretation of the mediation effects was determined (e.g., via statistical significance, magnitude of effect size) would clarify the importance of these findings.

The gender-specific analysis (Figure 2) provides interesting insights but could be expanded upon in the discussion. Gender differences in HRQoL and depression should be explored further in relation to physical activity.

discussion:

While the discussion addresses the core findings well, it would benefit from a deeper exploration of the clinical implications of the mediation results. For instance, what are the practical steps clinicians could take to reduce depression and improve HRQoL through PA interventions in this population?

The limitations are appropriately acknowledged, particularly regarding the cross-sectional nature of the study and the reliance on self-reported data. However, the potential for residual confounding should also be mentioned, as it is possible that other unmeasured factors may influence the relationship between PA, depression, and HRQoL.

conclusion:

The conclusion might benefit from a clearer call to action for future research. Given the cross-sectional design, a suggestion for longitudinal studies or interventions aimed at directly testing the effects of PA on depression and HRQoL would be valuable.

lastly:

Acknowledging that the data were collected during the COVID-19 pandemic and associated restrictions is important, as it could have influenced several variables in your study, such as physical activity levels, mental health (including depression), and overall quality of life.. limitations section (after line 338) you should include something regards that..

7. PLOS authors have the option to publish the peer review history of their article (what does this mean?). If published, this will include your full peer review and any attached files.

Reviewer #1: No

Reviewer #3: No

Reviewer #4: No

---

## [Editor Report · Acceptance letter]

21 Nov 2024

PONE-D-24-24159R1 

PLOS ONE

Dear Dr. Kim, 

I'm pleased to inform you that your manuscript has been deemed suitable for publication in PLOS ONE. Congratulations! Your manuscript is now being handed over to our production team.

Kind regards, 

on behalf of

Dr. Md. Feroz Kabir 

Academic Editor

PLOS ONE